# Transcriptomic Analysis of Spleen Revealed Mechanism of Dexamethasone-Induced Immune Suppression in Chicks

**DOI:** 10.3390/genes11050513

**Published:** 2020-05-06

**Authors:** Yujie Guo, Aru Su, Huihui Tian, Minxi Zhai, Wenting Li, Yadong Tian, Kui Li, Guirong Sun, Ruirui Jiang, Ruili Han, Fengbin Yan, Xiangtao Kang

**Affiliations:** 1College of Animal Science and Veterinary Medicine, Henan Agricultural University, Zhengzhou 450046, China; 15093351679@163.com (Y.G.); 18339915359@163.com (A.S.); thh2610@163.com (H.T.); l609558944@163.com (M.Z.); liwenting_5959@hotmail.com (W.L.); ydtian111@163.com (Y.T.); likui0420@sina.com (K.L.); grsun2000@126.com (G.S.); jrrcaas@163.com (R.J.); Rlhan@126.com (R.H.); 2Henan Innovative Engineering Research Center of Poultry Germplasm Resource, Zhengzhou 450046, China

**Keywords:** chicken, spleen, stress, immunosuppression, RNA-seq

## Abstract

Stress-induced immunosuppression is a common problem in the poultry industry, but the specific mechanism of its effect on the immune function of chicken has not been clarified. In this study, 7-day-old Gushi cocks were selected as subjects, and a stress-induced immunosuppression model was successfully established via daily injection of 2.0 mg/kg (body weight) dexamethasone. We characterized the spleen transcriptome in the control (B_S) and model (D_S) groups, and 515 significant differentially expressed genes (SDEGs) (Fragments Per Kilobase of transcript sequence per Millions base pairs sequenced (FPKM) > 1, adjusted *p*-value (padj) < 0.05 and Fold change (|FC|) ≥ 2) were identified. The cytokine-cytokine receptor interaction signaling pathway was identified as being highly activated during stress-induced immunosuppression, including the following SDEGs—*CXCL13L2, CSF3R, CSF2RB, CCR9, CCR10, IL1R1, IL8L1, IL8L2, GHR, KIT, OSMR, TNFRSF13B, TNFSF13B*, and *TGFBR2L*. At the same time, immune-related SDEGs including *CCR9*, *CCR10, DMB1, TNFRSF13B, TNFRSF13C* and *TNFSF13B* were significantly enriched in the intestinal immune network for the IgA production signaling pathway. The SDEG protein-protein interaction module analysis showed that *CXCR5, CCR8L, CCR9, CCR10, IL8L2, IL8L1, TNFSF13B, TNFRSF13B* and *TNFRSF13C* may play an important role in stress-induced immunosuppression. These findings provide a background for further research on stress-induced immunosuppression. Thus, we can better understand the molecular genetic mechanism of chicken stress-induced immunosuppression.

## 1. Introduction

Stress-induced immunosuppression refers to a condition in which the body is affected by stress factors, the immune system is underdeveloped and the immune organ cells and tissues are damaged, resulting in abnormal immune function and the temporary or persistent dysfunction of the immune response [1,2]. At present, stress-induced immunosuppression is one of the most widespread problems in intensive breeding of food animals, especially in poultry breeding, causing serious threats to animal product food safety and public health.

As an important steroid hormone, glucocorticoids are one of the main drug classes for the treatment of immune system diseases and malignant lymphoid tumors. Glucocorticoids are also one of the main inducers of lymphocyte apoptosis [3,4]. A long-term stress response can increase glucocorticoid secretion, causing immunosuppression and reducing the animal’s ability to resist disease [5]. Studies in humans and mice have shown that dexamethasone (Dex) [6,7], as a glucocorticoid with a strong inhibitory effect on immune function, can induce apoptosis of immature spleen cells and thymus cells and induce immune suppression. As the largest immune organ in poultry, the spleen plays an important role in regulating cellular and humoral immunity. When stress immunosuppression occurs, immune-related genes in spleen tissues are significantly affected. Therefore, it is of great importance to study the effect of stress-induced immunosuppression on splenic gene expression to clarify the mechanism of stress-induced immunosuppression.

In order to analyze the effects of stress-induced immunosuppression on the spleen immune function of chickens, we constructed a chicken stress-induced immunosuppression model by exogenous injection of 2.0 mg/kg (body weight) Dex, referring to previous studies [8,9], and analyzed the model group and control group chicken spleen transcriptome using RNA-Seq technology. This study will help in the understanding of the molecular mechanisms of chicken stress-induced immunosuppression, and the candidate genes screened may provide a basis for improvement of animal antistress abilities using genetic selection.

## 2. Materials and Methods

### 2.1. Ethics Statement

This animal study protocol was approved by the Animal Care Committee of the College of Animal Science and Veterinary Medicine, Henan Agricultural University (17-0126). Animals were euthanized with pentobarbitone sodium before tissue sampling. All efforts were made to comply with animal welfare guidelines and minimize animal suffering.

### 2.2. Experimental Animals and Sample Collection

A total of 180 7-day-old Gushi cocks with healthy, uniform body weight were selected from the Animal Center of Henan Agricultural University. The chicks were randomly divided into the control group (D_S group) and the experimental group (B_S group) with three replicates in each group and 30 chicks in each repeat. All groups were raised in separate, clean chicken houses. Their management was carried out in accordance with the chick management manual, but without routine immunization and beak trimming.

The D_S group was given subcutaneous injections of 2.0 mg/kg (body weight) Dex daily, while the B_S group was given subcutaneous injections of the same amount of normal saline daily as the control. Blood and spleen tissue samples were obtained from 9 chicks (3 in each repeat) in each group at 0, 3, 7 and 10 days after Dex administration. After 7 days of treatment, the stress-induced immunosuppression model was successfully constructed by detection of immune-related indicators, including physiological indexes, such as corticosterone (CORT), glucose (GLU), and total protein (TP) in the peripheral blood of the chickens; serum CD3^+^, CD4^+^, *IgG*, *TNF-α*, *IL-6*, and *IL-1β* (Appendix A, Appendix A). Three chickens were selected from each group, and the D_S group selected three chickens with the most severe thymic atrophy, and spleen tissues were collected immediately after euthanasia. After liquid nitrogen freezing, the tissue samples were stored at −80 °C for RNA extraction. 

### 2.3. RNA-Seq Library Construction and Sequencing

The total RNA of three biological replicate chicken spleen tissues from each group was extracted using an RNAiso Plus kit (Takara, Kyoto, Japan). RNA was detected free of contamination and degradation using 1% agarose gel, and the RNA purity was estimated using a NanoPhotometer^®^ spectrophotometer (IMPLEN, Westlake Village, CA, USA). RNA concentration was measured using an Qubit ^®^ RNA Assay Kit in Qubit^®^ 2.0 Flurometer (Life Technologies, Carlsbad, CA, USA), and RNA integrity was assessed using the RNA Nano 6000 Assay Kit of the Bioanalyzer 2100 system (Agilent Technologies, Santa Clara, CA, USA). The results showed that the RNA was intact and free of DNA contamination. After the samples were qualified, a total of 3 µg RNA from each sample was used as input material, and six RNA sequence libraries were constructed—D_S_1, D_S_2, D_S_3 and B_S_1, B_S_2 and B_S_3. The quality of the library was evaluated using the Agilent Bioanalyzer 2100 system. Specific operations were carried out in strict accordance with the NEBNext^®^ UltraTM RNA Library Prep Kit Instructions for Illumina^®^ (NEB, Ipswich, MA, USA).

The libraries were sequenced on a 2 × 150 nt Illumina Hiseq system and produced 150 bp paired-end reads. The raw data in the Fastq format were first processed by an internal script, then the raw reads of the sequencing were obtained by removing the low-quality readings, with an Qphred ≤ 20 clear reading > 50%. At the same time, the Q20, Q30 and GC contents were calculated in the clean reads. All downstream analyses were based on clean, high-quality data. The chicken genome assembly (Gallus Gallus 4.0) and gene model annotation files were downloaded from Ensemble [10,11], and the paired-end clean readings were matched to the reference genome using TopHat v2.0.12 [12]. Known and novel transcripts in the matching results were identified using a transcriptional assembly method based on Cufflinks v2.1.1 [13] reference annotation, and the readings were mapped to each gene using the HTSeq v0.6.1 count. The FPKM of each gene was then calculated based on the length of the gene, since FPKM also takes into account the effect of sequencing depth and gene length on readings.

### 2.4. Differential Expression Analysis 

Based on the FPKM values of the Illumina sequencing data, mRNA expression levels in six different libraries of the two groups were evaluated, and differential expression analysis was performed using the DESeq2 R v1.14.1package. The *p* value was adjusted using Benjamini and Hochberg to control FDR. In this study, FPKM > 1, adjusted *p* value (padj) < 0.05 and |FC| ≥ 2 were defined as significant differential expression genes (SDEGs). The heat map clustering analysis of SDEGs was performed using a pheatmap R package.

### 2.5. GO and KEGG Enrichment Analysis

The GOseq R package [14] was used to identify significantly enriched gene ontology (GO) terms of SDEGs. The KEGG (Kyoto Encyclopedia of Genes and Genomes) PATHWAY [15] is the main public pathway database for understanding the functions of genetic biology. We used KOBAS software (v2.0) to analyze the statistical enrichment of SDEGs in the KEGG PATHWAY to identify enrichment pathways and to elucidate group differences in cellular pathways. GO terms or KEGG pathways with a corrected *p*-value (*q* value) < 0.05 were considered to be significantly enriched.

### 2.6. qRT-PCR Analysis of SDEGs

To corroborate the RNA-Seq results, we randomly selected 10 genes from the SDEGs for qRT-PCR validation, including five upregulated genes and five down-regulated genes. Chicken spleen tissue total RNA was extracted, and a reverse transcription reaction was carried out using a primerScriptTM RT kit (Takara, Kyoto, Japan) to obtain cDNA for qRT-PCR. The reaction was carried out using the LightCycler^®^ 96 instrument qRT-PCR system (Roche, Basel, Switzerland) and SYBR^®^ PremixEx TaqTM kit (Takara, Kyoto, Japan). The qRT-PCR amplification procedure was as follows—95 °C for 3 min, 35 cycles of 95 °C for 30 s, 60 °C for 30 s, 72 °C for 30 s, and extended at 72 °C for 10 min. In addition, before the gene quantification, three common chicken housekeeping genes (B2M, β-actin and GAPDH) were tested [16] and evaluated using GeNorm [17]. Since GAPDH showed a lower M value and was stable in Gushi chickens [18], it was used as an internal reference gene. The relative expression changes of the genes were calculated using the 2^−^^ΔΔCt^ method. The qRT-PCR samples were the same as the RNA-seq samples, with three biological repeats for each group and three repetitions for each sample. The primer sequences are listed in (Appendix A).

### 2.7. Protein Protein Interaction Analysis 

Protein-protein interaction (PPI) analysis was based on the STRING database [19]. PPI network visualization was created with Cytoscape (version 3.6.1), and core proteins were identified by calculating the number of interactions between each network node.

### 2.8. Statistical Analysis 

The experimental data were presented as the mean ± standard deviation of three repetitions. GraphPad Prism (version 5.0) software (San Diego, CA, USA) was used for statistical analysis of the qRT-PCR graphs. One-way ANOVA and Student’s *t*-test were used for statistical analysis. *p* < 0.05 is considered to be statistically significant, and *p* < 0.01 is considered to be statistically highly significant. 

## 3. Results

### 3.1. RNA Deep Sequencing Information

Illumina HiSeq sequencing results showed that six libraries B_S_1, B_S_2, B_S_3, D_S_1, D_S_2 and D_S_3 from the two groups generated a total of 334 million raw reads. After removing the low-quality reads, 312 million clean reads were retained, accounting for 94.05%, of which 83.96% were mapped to the chicken genome. Among these total mapped reads, 64.1–67.8% were mapped to the exons, 23.1–25.8% were mapped to the intergenic regions, and only 8.8–10.2% were mapped to the introns. The GC content of the clean reads ranged from 48.54% to 50.41%. The raw sequence data were deposited in the NCBI database Sequence Read Archive with the accession number PRJNA577328, PRJNA623599. The specific results are shown in Table 1.

The correlation of gene expression level between samples is an important index of the reliability of the experiment and the reasonableness of the sample selection. The correlation analysis shows that the R2 between the biological duplicates in this study was higher than 0.92, as shown in Figure 1.

### 3.2. Differential Gene Expression in Spleen Tissue

Of a total of 19,262 genes detected in this study, 515 SDEGs were obtained from the D_S group and B_S group (FPKM > 1, padj <0.05 and |FC| ≥ 2), of which 281 genes were upregulated, 234 genes were down-regulated, 421 genes were known, and 94 genes were novel. Among the 515 SDEGs, major histocompatibility complex, class II, DM beta 1 (*DMB1*) was the most significantly upregulated gene with a log_2_FoldChange of 3.803, and microtubule associated protein 1 light chain 3 gamma (*MAP1LC3C*) was the most significantly down-regulated gene with a log_2_FoldChange of −3.8482. These results are shown in Figure 2 (Appendix A). Figure 3 shows the cluster analysis results of the SDEGs. It is clear that the gene expression patterns among the biological replicates are similar.

### 3.3. Gene Ontology (GO) Analysis

A total of 515 SDGEs were enriched for 4758 GO terms. We are concerned here with the top 30 GO terms, as shown in Figure 4 (Appendix A).

In the biological process, the primary subcategories are the immune system process, immune response, response to cytokine, chemokine-mediated signaling pathway, cellular response to cytokine stimulus, regulation of the immune system process and immune effector process. The immune system process was the most affected of the subcategories, including upregulated genes *CSF3R, GCNT1, MAPKAPK3, PDK4, PDE5A*, down-regulated genes *CCR10, OASL, RSAD2, SERPING1, THY1* and so on. In cell composition, the most affected subcategory was extracellular space. As for molecular function, the most affected subcategories were cytokine receptor activity (upregulated—*CCR8L, CCR9, CSF3R, GHR, IL1R1, IL1RL1, LPR, OSMR*, and down-regulated—*CXCR5, CCR10*), and CXCR chemokine receptor binding (down-regulated—*CXCL13L2, IL8L1, IL8L2*). These genes may play key regulatory roles in stress-induced immunosuppression.

### 3.4. KEGG Pathway Enrichment Analysis

KEGG pathway enrichment analysis was performed on the SDEGs to further understand the regulatory role played by SDEGs in stress-induced immunosuppression. 

We enriched 515 SDEGs into 91 KEGG pathways, of which the first 20 KEGG pathways are shown in Figure 5 (Appendix A). The immune-related pathways include cytokine-cytokine receptor interaction, intestinal immune network for IgA production, influenza A, Vascular endothelial growth factor (*VEGF*) signaling pathway, Toll-like receptor signaling pathway, RIG-I-like receptor signaling pathway and Jak-STAT signaling pathway. In the top three most significantly enriched pathways, the immune-related genes *CXCL13L2, CSF3R, CSF2RB, CCR9, CCR10, CCR8L, CXCR5, GHR, IL1R1, IL8L1, IL8L2, KIT, KDR, OSMR, TNFRSF13B, TNFSF13B, TNFRSF13C* and *TGFBR2L* were enriched in cytokine-cytokine receptor interaction, *TNFRSF13B, TNFRSF13C, TNFSF13B, DMB1, CCR9* and *CCR10* were enriched in intestinal immune network for IgA production, and *TLR7, STAT1, DMB1, RSAD2, Mx1, IL8L2, EIF2AK2, IFIH1, IL8L1 and KPNA2* were enriched in influenza A.

### 3.5. Corroboration of RNA-Seq Results by qRT-PCR

Ten genes (*CXCL13L, IL8L1, MX1, AVD, MAP1LC3C, UBIAD1, CCR9, NAALAD2, DMB1, PPYR1*) were randomly selected from the RNA-seq results, and were used as the classic representatives of stress affecting immune function. Expression of genes was corroborated by qRT-PCR. The results of the qRT-PCR showed that *CXCL13L, IL8L1, MX1, AVD, MAP1LC3C* and *UBIAD1* were significantly down-regulated in the D_S group compared with the B_S group (*p* < 0.05), while *CCR9, NAALAD2, DMB1* and *PPYR1* were significantly upregulated in the D_S group (*p* < 0.05) (Figure 6), which was consistent with the results of RNA-Seq. These results indicate that RNA-seq was a reliable reference method for expression profiling analysis and the sequence quality was acceptable. The FPKM values of the RNA-seq data are shown in Appendix A.

### 3.6. Protein Protein Interaction Analysis of SDEGs

The PPI network of the SDEGs consists of 104 proteins and 172 pairs of PPIs (Appendix A), of which the *ANXA1, CXCR5, CCR8L, CCR9, CCR10, GNGT2, IL8L2, IL8L1, LPAR1, MX1, OASL, PPYR1* and *SSTR3* genes have more than eight interactions (Appendix A). Eight modules including 37 proteins were obtained using the default criteria (Degree Cutoff: 2, K-Core: 2, Node Score Cutoff: 0.2, Max. Depth: 100) (Appendix A). Among these, we found that the first module contained the *ANXA1, CXCR5, CCR8L, CCR9, CCR10, GNGT2, IL8L2, IL8L1, LPAR1, PPYR1* and *SSTR3* genes, of which *CXCR5, CCR8L, CCR9, CCR10, IL8L2* and *IL8L1* were enriched in the cytokine-cytokine receptor interaction signaling pathway. The sixth module contained the *TNFRSF13B, TNFRSF13C* and *TNFSF13B* genes, of which *TNFRSF13B* and *TNFRSF13C* were enriched in the intestinal immune network for the IgA production signaling pathway (Figure 7).

## 4. Discussion

At present, the poultry meat and egg products industry is enjoying great success, as these products are considered a healthy alternative to red meat and other protein sources [20,21]. If this trend is to be maintained, solutions must be found to improve the resistance of chickens to disease, which is often weakened by stressful environments. Stress-induced immunosuppression is characterized by vaccination failure and increased disease morbidity and mortality in poultry [22,23,24,25]. Therefore, it is of great theoretical and practical importance to study the regulatory pathway and mechanism of stress-induced immunosuppression.

In poultry, there are many types of stressors that can induce immunosuppression and the consequences of immunosuppression from the same stressor can vary with the intensity of the response. Therefore, the establishment of corresponding immunosuppression models based on different types of stress is the basis for studying the mechanism of stress-induced immunosuppression. Dexamethasone is a stressor commonly used in laboratories to induce immunosuppression in poultry [26,27]. Vishwas et al. [28] used Dex to construct an immunosuppression model of golden hamsters, and the results showed that the spleen weight of the Dex-treated group was significantly lower than that of the control group, the serum cortisol level was significantly increased in the Dex-treated group, and the serum level of the pro-inflammatory cytokine IL-2 was significantly decreased in the Dex-treated group. In this study, the spleen weight of the D_S group was lower than that of the B_S group, but the difference was not significant. It may be caused by the young age of the experimental animals in this study. We made spleen tissue sections of the D_S group and B_S group, observed the spleen histology microstructure and found that the splenic corpuscle structure of the D_S group was not obvious and the number of them decreased (Appendix A). In addition, the thymus weight of the D_S group was significantly lower than that of the B_S group in this study (Appendix A), indicating that Dex treatment could cause thymus atrophy in chicks. Huff et al. [8] constructed a turkey immunosuppression model with intramuscular injection of 2 mg/kg weight Dex. Hebishima et al. [29] constructed a chicken stress-induced immunosuppression model by administering 10 μg/kg of Dex. In this study, 7-day-old Gushi cocks were selected as subjects, and a stress-induced immunosuppression model was successfully established with daily injection of 2.0 mg/kg (body weight) Dex.

We characterized the spleen transcriptome in B_S and D_S to detect key genes and pathways involved in stress-induced immunosuppression. A total of 19262 genes were detected in this study. Further analysis showed that 515 SDEGs were obtained from the D_S group and B_S group (FPKM > 1, padj < 0.05 and |FC| ≥ 2). *DMB1* (major histocompatibility complex, class II, DM beta 1) the most significantly upregulated gene, is part of the major histocompatibility complex II. The major histocompatibility complex (MHC) is a genetic region on a chromosome composed of tightly linked and highly polymorphic loci, which plays an important role in the immune response of animals [30,31]. The chicken class I system has been particularly well-studied, and in contrast, little is known about the chicken class II system and what it might tell us about evolution. The MHC has been shown to influence chicken resistance to Marek’s disease virus (MDV), and the MHC-I pathway genes exhibited higher transcripts after MDV infection. However, *DMB1* as nonclassical MHC-II genes exhibited lower expression in spleens of MDV-infected chickens [32]. Parker et al. [33] found that there may be two class II systems in chickens, and the *DMB1* gene is only highly expressed in the spleen and intestine. In this study, *DMB1* gene was strongly activated in the Dex-induced spleen of chicks, and we speculated that it might play an important role in the occurrence of stress-induced immunosuppression. It is imperative to study the specific mechanism of *DMB1* involved in stress-induced immunosuppression. Autophagy is a process of self-degradation of lysosome-dependent endogenous substrates in eukaryotic cells, by which the living body maintains the balance of protein metabolism and the stability of the intracellular environment [34]. This process plays an important role in cell waste removal, structure reconstruction and growth and development. *MAP-LC3* is an important gene encoding an autophagy-related protein and plays an essential role in the regulation and process of autophagy [35]. Chen et al. [36] found that H_2_O_2_-induced oxidative stress had a negative effect on broilers, and the *MAP1LC3* mRNA level of 2.96 mM/kg BW H_2_O_2_-treated broilers was significantly increased. As the most significantly down-regulated gene in this study, *MAP1LC3C* is worthy of attention. Autophagy is a homeostatic biological process [34]. We speculated that Dex-induced decreased expression of *MAP1LC3C* might inhibit the formation of autophagosomes and trigger the occurrence of immune response. It is necessary to carry out *MAP1LC3C* research on chicken immune cells to explore its specific regulatory mechanism.

A moderate immune response can maintain immune system homeostasis [37]. Long-term stress stimulation can damage the body’s immune system, induce immune response disorders, and lead to inflammation and high susceptibility to various diseases. In this study, the SDEGs were mainly concentrated in immune system processes—the immune response, response to cytokine, chemokine-mediated signaling pathway, cellular response to cytokine stimulus and other immune-related subcategories. We speculate that Dex can activate the spleen immune response. It is well-known that cytokines play an important role in regulating cell growth and differentiation in the body, and are involved in immune and inflammatory responses and wound healing [38,39,40]. According to their functions in the host defense response, cytokines can be divided into two categories—pro-inflammatory cytokines and anti-inflammatory cytokines [41]. In the process of the development of the inflammatory response, the body releases TNF-α, IL-1, IL-6 and other pro-inflammatory cytokines, and at the same time, it also secretes IL-10, IL-4 and other anti-inflammatory cytokines to fight the inflammatory response and maintain the stability of the body’s internal environment. Glucocorticoids are anti-inflammatory molecules in the traditional view; however, emerging evidence suggests that the role of glucocorticoids is more complex than anticipated [42]. Glucocorticoids have been shown to down-regulate the expression of various pro-inflammatory factors such as IL-1β, IL-6, IL-8, TNF-α, and upregulate the expression of anti-inflammatory genes, including lipocortin-1, IL-10, and IL-1R antagonist [43]. The cytokine-cytokine receptor interaction signaling pathway was identified as being highly activated during stress-induced immunosuppression in this study. Among them, *CXCR5, CCR10, CXCL13L2, IL8L1, IL8L2* and other pro-inflammatory factors and chemokines were down-regulated, suggesting that Dex may inhibit the immune response of chicken spleen tissues, leading to the occurrence of stress-induced immunosuppression. These genes may play key regulatory roles in stress-induced immunosuppression. In addition, *CCR9, CCR8L*, colony stimulating factor 3 receptor (*CSF3R*), and *IL-1R* were significantly upregulated, probably because after Dex treatment, the body stimulated the production of related chemokines and cytokines in response to inflammatory stimuli in order to maintain homeostasis.

We also demonstrated the protein interaction network of the SDGEs (Appendix A). Among them, the *IL8L2, IL8L1, LPAR1, GNGT2, ANXA1, CXCR5, CCR8L* and *MX1* genes are the core nodes in PPI with more than eight interactions. In order to further identify the protein interaction module that plays an important role in stress-induced immunosuppression, we conducted PPI module analysis. The module analysis showed that *IL8L2, IL8L1, CXCR5, CCR8L, CCR9* and *CCR10* may play an important role via the cytokine-cytokine receptor interaction signaling pathway, and *TNFRSF13B* and *TNFRSF13C* may play an important role via the intestinal immune network for the IgA production signaling pathway. It is reported that TNFRSF13B is a type I transmembrane glycoprotein and is a tumor necrosis factor receptor [44,45]. *TNFRSF13B* and *TNFRSF13C* combine with *TNFSF13B* to form a trimer [46,47,48], which initiates signal transduction and participates in complex immune responses and pathological damage processes. This *TNFSF13B/TNFRSF13C* system is crucial for B cell maturation and survival. Human studies show that inhibition of *TNFSF13B/TNFRSF13C* signaling may be a therapeutic option for the treatment of B-cell-mediated autoimmune diseases [49,50]. Data from clinical trials has proved that blocking *TNFSF13B* via blocking reagents is an effective approach for some autoimmune diseases [51]. Mohd et al. [52] found that the expression level of *TNFSF13B* in infectious bursal disease virus (IBDV)-infected chickens was significantly decreased. Dulwich et al. [53] showed that TNFSF13B also showed significantly low expression in IBDV-infected B cells in the bursae of Fabricius, causing speculation that *TNFSF13B* may contribute to IBDV-mediated immunosuppression. The data from this study show that *TNFRSF13B, TNFRSF13C* and *TNFSF13B* were down-regulated in the spleen tissues of the stress-induced immunosuppression model, which is consistent with the data from the above immune-related diseases. Therefore, we speculate that *TNFRSF13B, TNFRSF13C* and *TNFSF13B* may play an important role in stress-induced immunosuppression.

There are limitations in the model compared to the natural stress condition. The candidate genes in this study should be eventually validated under different natural stress conditions in further studies to finally find the most extensive candidate genes related to most stressors.

## 5. Conclusions

In summary, we described the transcriptome profiles of a dexamethasone-induced model of chicken stress-induced immunosuppression, and identified 515 SDEGs. The cytokine-cytokine receptor interaction signaling pathway was identified as being strongly activated during stress-induced immunosuppression. Through GO, KEGG, and PPI analysis, we found that *IL8L2, IL8L1, CXCR5, CCR8L, CCR9* and *CCR10* may play an important role in stress-induced immunosuppression via the cytokine-cytokine receptor interaction signaling pathway, and that *TNFRSF13B* and *TNFRSF13C* may play an important role in stress-induced immunosuppression via the cytokine-cytokine receptor interaction signaling pathway and the intestinal immune network for the IgA production signaling pathway. These results provide a valuable basis for further study of the molecular mechanism of stress-induced immunosuppression, as well as new insights for better understanding of the interactions among various factors related to stress-induced immunosuppression. Further research should validate the functions of these key interaction networks in chicken stress-induced immunosuppression at the cellular level.

## Figures and Tables

**Figure 1 genes-11-00513-f001:**
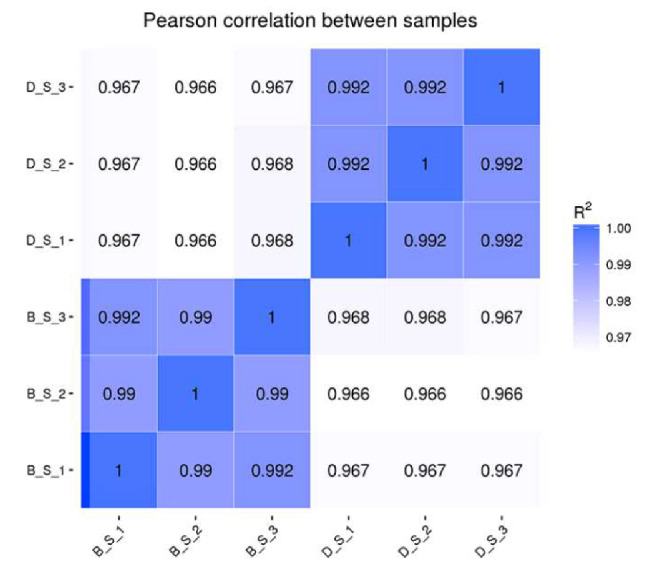
Pearson correlation between samples.

**Figure 2 genes-11-00513-f002:**
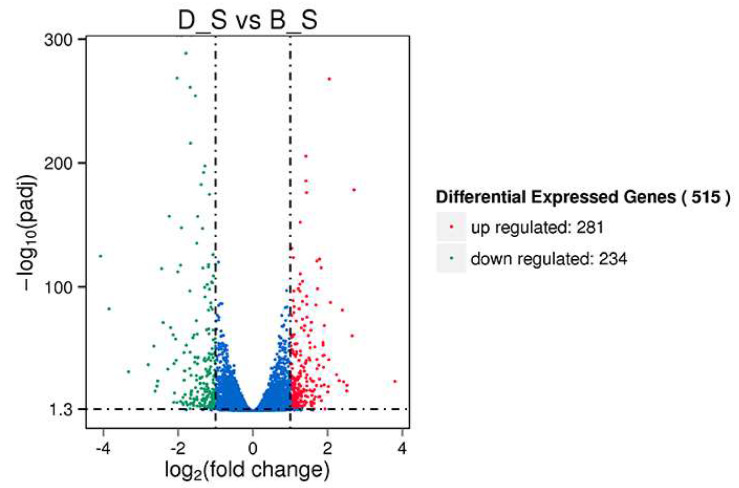
The volcano map depicts differentially expressed genes between the two groups of spleens. Significant differential expression genes (SDEGs) are shown as blue (down) and red (up) dots, and gray dots show a lack of significance.

**Figure 3 genes-11-00513-f003:**
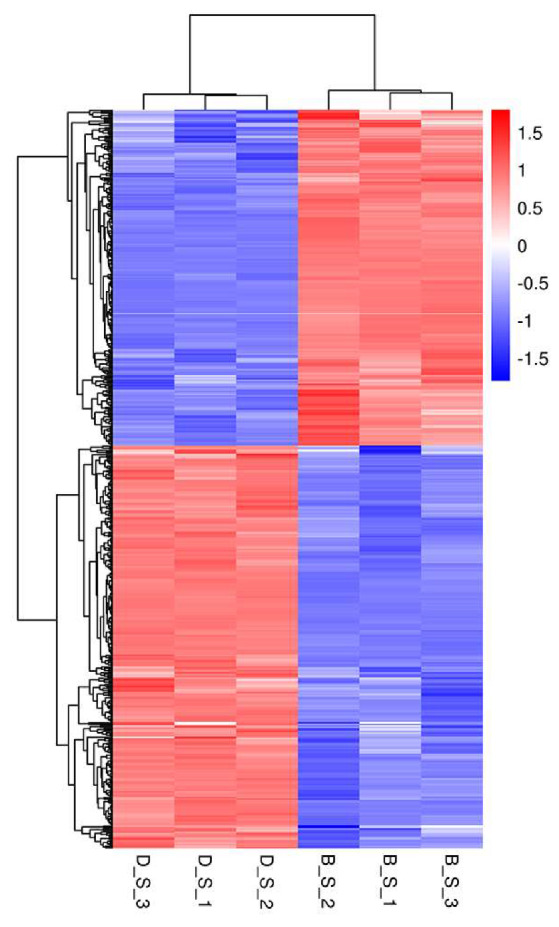
Cluster analysis of SDEGs in spleens between D_S group and B_S group by the FPKM value. High expression genes are shown in red and low expression genes are shown in green. The closer the branches of the two samples are, the closer the expression patterns of all genes in the two samples are, and the closer the trend of gene expression are.

**Figure 4 genes-11-00513-f004:**
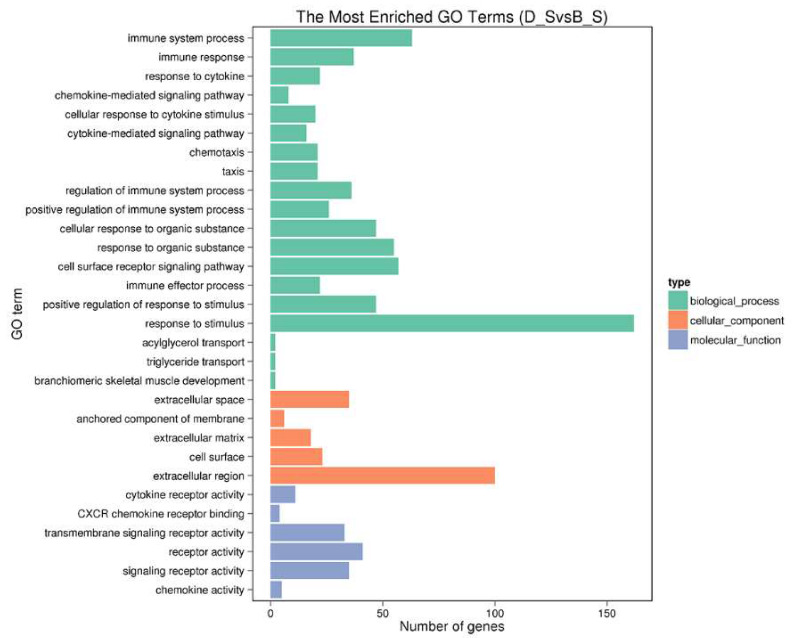
The top 30 enriched gene ontology (GO) terms of SDEGs. The X-axis shows the SDEG number of each GO term; the Y-axis shows the SDEG-enriched GO term.

**Figure 5 genes-11-00513-f005:**
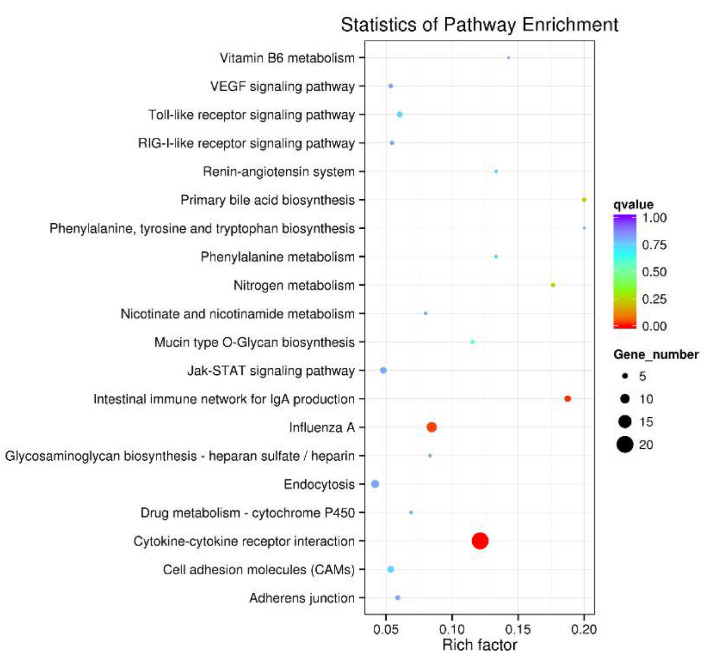
The top 20 Kyoto Encyclopedia of Genes and Genomes (KEGG) pathways of SDEGs of D_S group and B_S group. The color of the dot represents the *q* value, and the size of the dot represents the number of SDEGs enriched in the reference pathway.

**Figure 6 genes-11-00513-f006:**
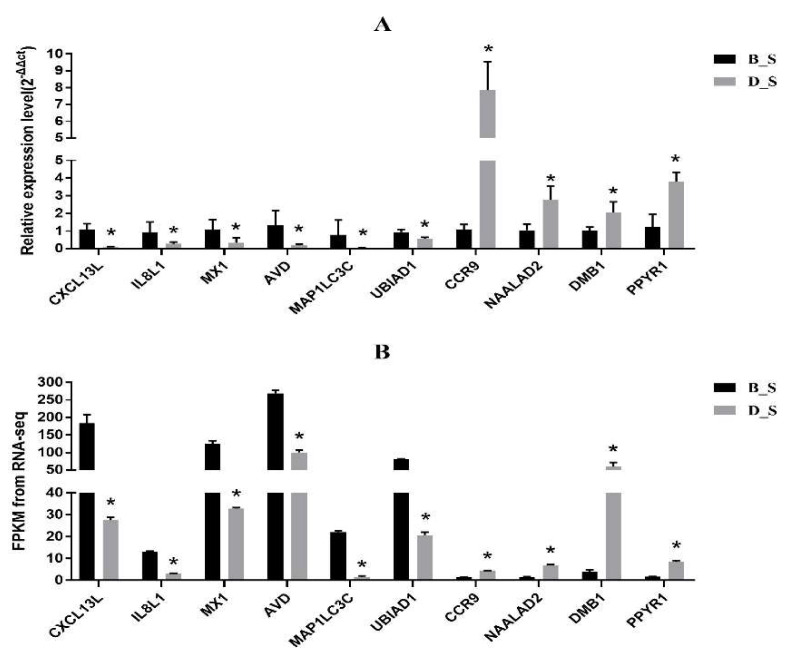
qRT-PCR corroboration of RNA-seq results. (**A**) The qRT-PCR result. (**B**) The RNA-seq result. The black column is the B_S group, and the gray column is the D_S group, * indicates significant difference (*p* < 0.05).

**Figure 7 genes-11-00513-f007:**
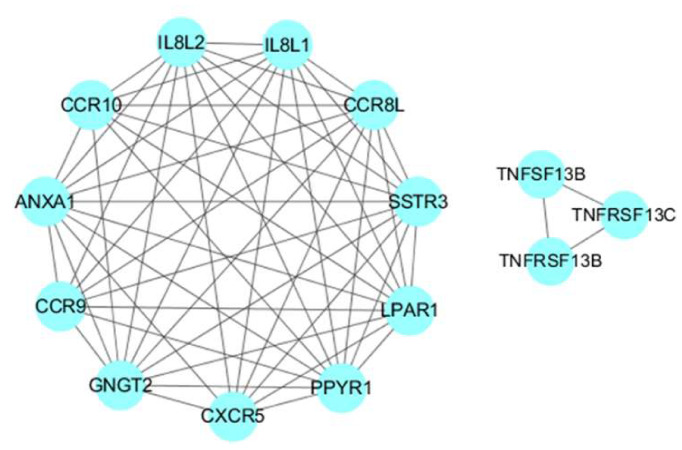
Important modules of the protein-protein interaction (PPI) network. The connection between proteins indicated that they interact and the confidence interval was greater than 0.7.

**Table 1 genes-11-00513-t001:** Characteristics of the reads from spleen libraries obtained from 2 groups.

Sample Name	Raw Reads	Clean Reads	Total Mapped	Multiple Mapped ^1^	Uniquely Mapped ^2^	Exo*n*%	Intergenic%	Intron%	GC Content (%)
B_S_1	49,138,008	44,665,444	37,779,488 (84.58%)	888,893 (1.99%)	36,890,595 (82.59%)	67.5	23.5	9.0	49.55
B_S_2	50,924,048	48,075,590	39,116,591 (81.36%)	943,790 (1.96%)	38,172,801 (79.40%)	67.8	23.4	8.8	50.41
B_S_3	57,824,730	55,293,336	45,753,593 (82.75%)	1,112,575 (2.01%)	44,641,018 (80.73%)	67.7	23.1	9.2	49.31
D_S_1	58,751,020	52,244,762	44,495,796 (85.17%)	1,004,681 (1.92%)	43,491,115 (83.24%)	64.1	25.8	10.1	48.54
D_S_2	59,055,618	56,786,086	47,986,552 (84.5%)	1,109,279 (1.95%)	46,877,273 (82.55%)	65.7	24.6	9.7	48.76
D_S_3	58,552,874	57,292,842	48,792,672 (85.16%)	1,116,745 (1.95%)	47,675,927 (83.21%)	65.0	24.9	10.2	49.08

^1^ Multiple mapped = number of clean reads and the ratio that matched two or more positions in the genome. ^2^ Uniquely mapped = number of clean reads and the ratio that matched only one position in the genome.

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
