# Peer review of "Transcriptomic Analysis of Spleen Revealed Mechanism of Dexamethasone-Induced Immune Suppression in Chicks"

_genes, 2020, doi:10.3390/genes11050513_

Round 1

Reviewer 1 Report

Authors report their research findings regarding transcriptomic response of thymus tissues after a single dose injection of Dexamethasone in chicken after 7 days.

Here are the major issues and questions regarding this manuscript.

  1. The authors only provide the minimum data regarding the dose and time point in this study (line 76). Why single dose 2.0 mg/kg dexamethasone after 7 days of treatment is the best approach to study immune suppression in chicks? There are not enough literatures to support the experimental design is appropriate in materials and methods. The authors quoted their previous work (line 77). Unfortunately, no data is provided to resolve this issue in this publication. Even providing the positive response of physiological indexes (line 76, Table S1), how do you know these indexes reflect the optimum physiological response of immune suppression? Time points with different doses response studies would be the best approach to make sure the optimum physiological of immune suppression is reached before further decision on transcriptomic study of immune suppression.

  1. Total sample sizes (180) is large enough for the injections from two groups with control (BS) = 90 and experimental (DS) = 90. However, the final samples chosen for RNAseq study are very small (line84-85). Without any physiological indexes provided for these selected individuals (BS = 3 and DS = 3), these selected samples may not be statistically sound to reflect the original samples which response positively to immune suppression after Dexamethasone injection. Since the standard variation is quite high in some of markers such as IL-6 (see Table S1: BS = 32.16±6.73; DS = 39.95±3.13), I suggest authors should provide more details regarding the physiological indexes for the six selected samples. On the other hand, with only six samples chosen for further RNA purification, there is no data to mention the quality of RNA (line 81-83). Is there any gel image or RIN values to show the quality of RNA?

  1. There are two major issues concerning the RNA deep sequencing data. I suggest the authors should deposit their RNA-seq data either in the GEO database or in the NCBI’s Sequence Read Archive for reviewers to examine the data quality and for further usage. Secondly, gene annotation is the key success for the data analysis of the RNA sequencing and I suggest to use the most updated reference genome (either Gallus_gallus-5.0 or GRCg6a) instead of the oldest version in the manuscript (line 93). Source of the reference genome from Ensemble should be mentioned and provided with the website address (line 94 no address mentioned).

  1. There are several problems with validation of SDEGs. There are 515 SGEGs found in this study validation all of them using qRT-PCR is not necessary. However, there are only 14 signature genes correspond to the key pathways mentioned in this study (Line 22 to 28). I highly recommend that the authors need to validate all of them, to strength all the pathways that are involved in immune suppression response hypothesis, evidentially by qRT-PCR instead of picking target genes randomly for validation. Using qRT-PCR tool in this study, there is no cited reference regarding that using the chicken GAPDH gene (line 123) is appropriate as an internal gene without providing their own data for validation. Also, providing accurate PCR primers information is important for other researchers using the same genes for PCR purpose. The obvious mistake should be correct (line 125, Table S1 should be replaced by Table S2). In table S2, amplicon sizes and nucleotide sequence ID such as GenBank accession number for each gene are missing. Finally, there is a flaw regarding the qRT-PCR result (line 207-209). Based on figure 6, UBIAD1 is not up-regulated and should not be consistent with the result of RNA-seq.

Author Response

Thank you very much for working our paper. 

Reviewer 2 Report

This is a very well planned and executed study.  I found the methodology to be sound and the results very well presented.  My chief concern with this paper is that like many RNA-seq studies it presents a wealth of results, but offers little biological insight.

Specific points to address:

Can you describe how the dosage of 2.0 mg/kg was determined as appropriate in your model? Citation?

Besides the biochemical measures was any phenotypic data collected on the spleens? Weight?

qRT-PCR.  This approach is not a validation.  You are using an alternate approach (analog) to corroborate your digital data.  Even if the two approaches were more similar, the qRT experiment would have needed to use the same cDNA samples as used for sequencing.

The discussion needs to provide a better presentation on what has been elucidated by RNAseq.  What makes sense to the biology of the chicken immune response and what is surprising?  It is not surprising to see cytokines and their receptors affected, is there a distinction between pro- and anti- inflammatory molecules? How do the expression changes observed in this model system relate to what is known about true stress induced immunosuppression. 

Although lists are generated of the genes involved, there is little discussion about what the individual SDEGs actually do and how this relates to stress/immunosuppresion.

Please update your references regarding the structure and function of the avian MHC locus.

Other minor comments/suggestions are noted in the attached marked copy.

Author Response

(The authors gave the same response as above.)

Round 2

Reviewer 1 Report

Comments to your response letter as listed in details below:

From point 1:

Line 47-50: Any reasons try to avoid presenting the time points graphic data of CORT in your revised manuscript if this is one of the main selection criteria for D7 samples (N=3). What about the similar time points data for IL-6 and other indexes, are they available?

Line 50-54: D7 and D10 are both significant, why not do both or combined two time points samples for sequencing?

Line 73: No data is mentioned or present in this reference [10] regarding the thymus atrophy.

From point 2:

Line 122-124: Again this is not the correct reference and nothing is mentioned regrading thymus atrophy or protein sequencing.

Line 125-128: This study supposed to be an hypothesis driven and the main objection is to find differential expressed genes or markers after treatment of Dex in spleen. Using sequencing profile similarity among D_S samples (N=3) to assume that the original treatment is correct is correct approach and not scientific. 

Line 137-142: Visually, there is no obvious different between A & B. Any quantify values to show they are significantly different. 

From point 3:

Line 189: No items are available from NCBI for reviewer's validation. 

From point 4:

Line 272: Why beta actin is not chosen instead of GAPDH? They have the exactly same M value.

Author Response

Thank you very much for your comments and suggestions.

Reviewer 2 Report

The authors have made a great effort in improving this manuscript.  I believe all of my previous comments have now been addressed, and the discussion is much improved. 

I found 2 typos for correction 

Line 154 – fix typo, should read “raw sequence data”

Line 264 – fix typo, should read “We made spleen”

Author Response

Dear editor,

Thank you so much for your comments. We greatly admire your seriousness, responsibility and meticulous spirit. According to your comments, we have revised them:

“The raw sequence data” (line 153)

“We made spleen” (line 263)

We appreciate for Editors/Reviewers’ warm work earnestly, and hope that the correction will meet with approval. Once again, thank you very much for your comments and suggestions.

Sincerely,

Fengbin Yan